# Isochoric Supercooling Organ Preservation System

**DOI:** 10.3390/bioengineering10080934

**Published:** 2023-08-07

**Authors:** Gabriel Năstase, Florin Botea, George-Andrei Beșchea, Ștefan-Ioan Câmpean, Alexandru Barcu, Ion Neacșu, Vlad Herlea, Irinel Popescu, Tammy T. Chang, Boris Rubinsky, Alexandru Șerban

**Affiliations:** 1Department of Building Services, Faculty of Civil Engineering, Transilvania University of Brasov, 500152 Brasov, Romania; george.beschea@unitbv.ro (G.-A.B.); stefan.campean@unitbv.ro (Ș.-I.C.); 2Center of Excellence in Translational Medicine CEMT, “Dan Setlacec” Center of General Surgery and Liver Transplantation, Fundeni Clinical Institute, 022328 Bucharest, Romania; alexbarcu@gmail.com (A.B.); herlea2002@yahoo.com (V.H.); irinel.popescu220@gmail.com (I.P.); 3Department of Medical-Surgical and Profilactical Disciplines, “Titu Maiorescu” University, 040441 Bucharest, Romania; 4Criomec SA, 800219 Galati, Romania; ion.neacsu@criomecsa.ro; 5Department of Surgery, University of California, San Francisco, CA 94143, USA; tammy.chang@ucsf.edu; 6Department of Mechanical Engineering, University of California, Berkeley, CA 94720, USA; rubinsky@me.berkeley.edu; 7Department of Thermotechnics, Engines, Thermal and Refrigeration Equipment, Faculty of Mechanical Engineering and Mechatronics, University Politehnica of Bucharest, 060042 Bucharest, Romania; alexandru.serban@criomecsa.ro

**Keywords:** isochoric system, subfreezing temperatures, large organ preservation, cryopreservation, supercooling, liver, porcine/pig model

## Abstract

This technical paper introduces a novel organ preservation system based on isochoric (constant volume) supercooling. The system is designed to enhance the stability of the metastable supercooling state, offering potential long-term preservation of large biological organs at subfreezing temperatures without the need for cryoprotectant additives. Detailed technical designs and usage protocols are provided for researchers interested in exploring this field. The paper also presents a control system based on the thermodynamics of isochoric freezing, utilizing pressure monitoring for process control. Sham experiments were performed using whole pig liver sourced from a local food supplier to evaluate the system’s ability to sustain supercooling without ice nucleation for extended periods. The results demonstrated sustained supercooling without ice nucleation in pig liver tissue for 24 and 48 h. These findings suggest the potential of this technology for large-volume, cryoprotectant-free organ preservation with real-time control over the preservation process. The simplicity of the isochoric supercooling device and the design details provided in the paper are expected to serve as encouragement for other researchers in the field to pursue further research on isochoric supercooling. However, final evidence that these preserved organs can be successfully transplanted is still lacking.

## 1. Introduction

The objective of this technical paper is to introduce the design and usage protocol of a cold storage organ preservation system based on isochoric (constant volume) supercooling principles. Organ preservation plays a vital role in the success of organ transplantation by enabling the transportation and storage of organs from donors to recipients [1,2,3]. Currently, static cold storage is the standard method for organ preservation. Cold storage, commonly used for organ preservation, effectively slows down metabolic processes, reduces the demand for oxygen and nutrients, and lowers cellular metabolism by cooling the organ to around 4 degrees Celsius. While 4 °C cold storage is effective for short-term preservation, it has limitations regarding the duration of organ viability. Organs can only be stored for a limited time before their functionality begins to decline. The storage time for organs using cold storage can vary depending on the organ type and the specific circumstances. Here are the approximate storage times for commonly transplanted organs:Heart: The heart can typically be stored for about 4 to 6 h using cold storage.Liver: The liver can be stored using cold storage for approximately 8 to 12 h.Kidneys: Kidneys are relatively robust organs and can be stored using cold storage for a longer period compared to other organs. Typically, kidneys can be preserved for around 24 to 36 h using cold storage.Pancreas: The pancreas can be stored for approximately 12 to 18 h using cold storage.

To address these limitations, researchers and transplant professionals are actively exploring and developing new organ preservation techniques. Machine perfusion, originally conceived before cold storage, is currently garnering increased interest as an effective technique for preserving organs [4,5]. Cold storage with preservation solutions, first introduced for the kidney by Collins [6] and later improved for the liver by Southard and Belzer [7,8,9], became the gold standard due to the simplicity of the preservation process compared to machine perfusion. Cold storage involves flushing the procured organ with preservation solution at 0–4 °C, then immersing it into preservation solution at the same temperature until transplantation. The hypothermic environment is responsible for decreasing cellular metabolism [7]. In contrast, machine perfusion for the preservation of organs is much more complicated than cold storage. It involves pumping a preservation solution through the blood vessels of the organ to maintain its viability outside the body under continuous control of temperature and solution composition. Machine perfusion requires several steps, all of which are critical to the survival of the preserved organ. The steps required after harvesting of the organ include the following: (a) cannulation—the blood vessels of the organ, such as the artery and vein, are carefully cannulated; (b) connection to perfusion machine—the cannulated organ is then connected to a perfusion machine, which controls the flow of the preservation solution into and out of the organ’s blood vessels; (c) perfusion solution—the perfusion machine pumps a specially formulated preservation solution through the organ. This solution contains nutrients, oxygen, and other additives that help to maintain the organ’s cellular integrity and function during the preservation period; (d) oxygenation and nutrient supply—the perfusion solution provides the organ with a continuous supply of oxygen and nutrients, which helps to prevent cellular damage and deterioration during the preservation process; (e) temperature control—the perfusion machine can regulate the temperature of the preservation solution to match the optimal temperature for the specific organ being preserved. This temperature control helps to slow down the organ’s metabolic activity and reduces the risk of damage; (f) monitoring and assessment—throughout the perfusion process, the organ is closely monitored to assess its function and viability. Parameters like oxygen consumption, pH levels, and other indicators are checked to ensure the organ’s health. Comparing the steps necessary for cold storage with those needed for machine perfusion, it should be obvious why, with the introduction of cold storage solutions in the 1960s, cold storage became the preferable method for organ preservation over machine perfusion. However, as shown in the previous paragraph, the time of preservation with cold storage is limited. The hope that machine perfusion can extend the preservation time has triggered renewed interest in machine perfusion. Indeed, machine perfusion has successfully extended the preservation time over conventional cold storage at 4 °C, demonstrating also the potential to recondition marginal or extended criteria organs to make them suitable for transplantation [4]. This is why, despite the rather complicated machine perfusion preservation technique in comparison to the simple cold storage technique at 4 °C, machine perfusion is gaining renewed interest. However, finding ways to extend the period of cold storage for durations comparable to those achieved by machine perfusion would make cold storage competitive once again. This is because of the simplicity of the cold storage procedure.

Metabolism is known to be temperature-dependent [7,8]. One approach to extend ex corporis organ survival is to further reduce the metabolism of stored organs by lowering the storage temperature from 4 °C to temperatures below freezing. However, freezing by itself can lead to cell damage [10,11]. Extensive research on cell preservation through freezing has achieved success. It has been discovered that cells can survive freezing and be preserved at cryogenic temperatures by adding certain chemical species known as cryoprotectants, which protect them from ice-induced damage [12,13,14,15]. The ability to cryopreserve cells has had a profound impact on various fields, including medicine and biotechnology [16,17,18]. However, despite a century of research on organ preservation through freezing in a manner similar to cryopreservation of cells, no evidence of success has been found [2,19]. Consequently, new techniques are currently being developed for organ preservation at subfreezing temperatures without the presence of ice.

Preservation in a supercooled state at subfreezing temperatures is a possible way to reduce the metabolism of biological matter to subfreezing temperatures without the detrimental formation of ice. There is growing interest in preserving biological matter in a supercooled state for various applications in life sciences and food sciences. The idea of preserving biological matter in a supercooled state is not new [20]. A variety of techniques for supercooling preservation without ice were recently developed [21,22,23,24,25,26,27]. To list a few: perfusion with cryoprotectant solutions at temperatures at which they do not freeze [23], partial freezing to mimic survival of freeze-tolerant species [28], and deep-supercooling of large volumes with surface sealing with immiscible fluids [21]. A series of recent publications report the preservation of the liver in a supercooled state at subfreezing temperatures for extended periods of time, including week-long preservation [27,29]. The technology and the protocol for these was reported in [26]. Most of these technologies require the use of chemical additives, in one form or another, and they were all performed under atmospheric isobaric (constant pressure) conditions, which is the natural state in which life exists. Supercooling by mechanically generated hyperbaric pressure was also reported in [30,31,32], and that by electromagnetic means was reported in [33]. These systems are also isobaric. This work focuses on a novel technology for the preservation of biological matter through supercooling, specifically isochoric (constant volume) supercooling.

The first paper on the thermodynamics of isochoric (constant volume) cryopreservation was published in 2005 [34]. An advanced treatment of the process of isochoric freezing, employing the Helmholtz equilibrium, was published in [35]. The main features of isochoric freezing are explained in a simplified manner in Figure 1. The process of freezing in an isochoric system is shown in the schematic in Figure 1A. The figure illustrates that, in a chamber with rigid walls, ice nucleation begins at a nucleation site. Due to ice having a lower density than water, the pressure will increase in the constant volume chamber. Therefore, the thermodynamic path that this system will take, as the temperature of the system decreases, is along the liquidus line on the phase diagram in Figure 1B. Along this liquidus line, ice and water coexist in thermodynamic equilibrium; thus, the temperature uniquely specifies the pressure, and vice versa. Figure 1B illustrates the freezing process in the isochoric system as the temperature is reduced. Part of the water in the system will freeze, but the constraints of thermodynamic equilibrium require the presence of a two-phase solution—ice and water—to the triple point. The outcome of this constraint is illustrated by Figure 1C, which shows that, throughout the freezing process, until the triple point temperature of approximately −22 °C, there is a substantial percentage of unfrozen water in the system. As Figure 1A shows, when the biological matter to be preserved is placed in the portion of the chamber that will not freeze, the matter can reach substantial subfreezing temperatures without freezing. Although a similar effect could be attained through mechanically increasing the pressure, isochoric freezing offers the advantage of enforcing precisely defined conditions through equilibrium thermodynamics. In contrast to high-pressure isobaric systems, isochoric freezing only necessitates temperature control rather than simultaneous control of both temperature and pressure. This principle serves as the fundamental basis for the design of the isochoric chamber discussed in this paper. Figure 1B also illustrates the thermodynamic path in an isobaric system. In an isobaric system, at any pressure, when the temperature is reduced, the entire volume will freeze completely at the intersection between the constant pressure line and the liquidus line.

Isochoric freezing was successfully used for the preservation of biological matter and is generating interest [36,37,38]. In addition to isochoric freezing, theoretical and experimental studies have demonstrated that maintaining isochoric conditions enhances the stability of the metastable state of supercooling [39]. Recently, the utilization of isochoric supercooling has proven successful in preserving microchips with cells [40]. This technical paper provides comprehensive information on the principles underlying the design of an isochoric chamber intended for the preservation of large biological organs via isochoric supercooling. It provides detailed design specifications and usage guidelines, serving as a valuable resource for other researchers seeking to pursue further investigations in this emerging field.

## 2. Materials and Methods

### 2.1. Principles of Design

Isochoric freezing is characterized by the formation of ice, leading to a significant increase in pressure. These pressures can be substantial. Figure 1B illustrates that, at −10 °C, the pressure reaches approximately 100 MPa while, at −20 °C, it reaches around 200 MPa. When dealing with large organs like the liver, it becomes necessary to construct robust isochoric chambers capable of withstanding these high pressures.

In contrast, isochoric supercooling allows for maintaining atmospheric pressures inside the chambers, removing the size limitations for the chambers themselves. However, even under isochoric conditions, the supercooled system remains vulnerable to random nucleation. Since the chambers are designed to handle low pressures, a sudden increase in pressure can cause them to breach, resulting in the instantaneous freezing of the entire contents within the isochoric chamber. To preserve the integrity of the chamber, installing a pressure release valve can be considered. However, as a consequence, the entire contents of the isochoric chamber would freeze. Therefore, a critical aspect of the design of the isochoric supercooling system discussed in this paper is the inclusion of a control system that halts freezing before the pressure reaches the critical yield pressure of the chamber and initiates warming. The purpose of this control system is to salvage the preserved biological matter.

The isochoric supercooling system in this study consists of several key elements that will be discussed separately: the isochoric chamber, the refrigeration system, and the control system. The central component of the isochoric supercooling organ preservation system is the “isochoric chamber”, designed to accommodate various biological materials with varying volumes. Figure 2 presents a schematic representation of this concept. To achieve isochoric conditions, the isochoric chamber is completely filled with an aqueous solution with care to minimize the amount of free air in the chamber.

We found in earlier studies that the walls of the isochoric chambers provide ice nucleation sites that are very difficult to eliminate. To avoid surface treatment of large isochoric chambers, designed for multiple uses, the biological matter is stored in a completely sealed thin plastic bag, which transfers pressure and heat but not mass. In this study, we have used low-density polyethylene (LDPE) bags. LDPE is a commonly used commercial hydrophobic polymer that exhibits low surface energy and water repellency. Low-density polyethylene (LDPE) surfaces have been studied for their ability to inhibit ice nucleation [41].

The static cold storage of organs with isochoric supercooling follows a process akin to conventional cold storage protocols. In both methods, the organ undergoes perfusion with a low-temperature preservation solution specifically designed for that particular organ. This preservation solution achieves isotonic equilibrium with the organ, and may include options like Belzer UW^®^ (Bridge to Life (Europe) Ltd, London, UK) cold storage solution or HTK Custodiol^®^ (Essential Pharmaceuticals, LLC, Durham, NC, USA) solution.

Both conventional static cold storage and isochoric supercooling storage involve initiating the procedure by flushing the organ with the same isotonic cold storage preservation solution (e.g., UW or Custodiol^®^). Subsequently, in isochoric supercooling storage, the organ is placed inside a polymer bag filled with the conventional preservation solution used in static cold storage (e.g., UW or Custodiol^®^). This placement process is similar to how the organ is positioned in a sterile bag filled with the cold storage solution during conventional static cold storage. The only difference between the two methods lies in the type of bag used and the importance of minimizing the presence of air in the isochoric storage sealed bag, as discussed earlier. It is worth noting that avoiding air in the isochoric polymer bag is a simple task, achieved by overfilling the bag with the storage solution and using a zip lock mechanism for sealing. Further details of this process can be found in the Section 3. It is valuable to mention that handling the organ before and after isochoric supercooling cold storage mirrors the procedures of conventional static cold storage, presenting an equivalent level of difficulty.

During storage, the organ enclosed in the sealed polymer bag is submerged in the “isochoric chamber fluid”, which fills the isochoric chamber, as depicted in Figure 2. The formulation of the “isochoric chamber fluid” is intentionally designed to prevent freezing at the supercooled temperature used for storing biological matter. This particular composition ensures that the walls of the isochoric chamber do not induce nucleation, maintaining the supercooled state. It is crucial to highlight that the composition of the “isochoric chamber fluid” is carefully selected to remain non-freezing at the storage temperature. However, in contrast, the fluid within the polymer bag containing the organ has the same isotonic composition as the organ. Consequently, at the storage temperature, both the solution in the polymer bag and the organ exist in a metastable thermodynamically supercooled state and are susceptible to freezing.

Supercooling represents a metastable thermodynamic state, making ice nucleation a random and stochastic phenomenon. Several factors influence the probability of ice nucleation. One such factor is homogeneous nucleation, typically occurring at temperatures below the cold storage temperature intended for this application. Our research has demonstrated that isochoric conditions further lower the temperature of homogeneous nucleation [42], effectively eliminating the probability of this type of nucleation in this system.

Another mechanism of ice nucleation is surface nucleation caused by container roughness and composition. In our design, the use of the “isochoric chamber fluid” eliminates the impact of the isochoric chamber wall roughness on surface nucleation, and the hydrophobic polymer bag minimizes the effect of wall composition. Consequently, the remaining nucleation mechanism arises from random nuclei in the bulk of the solution in the polymer bag. Nonetheless, our previous studies have shown that confinement within an isochoric chamber can stabilize the metastable supercooled state [39].

It is essential to emphasize that isochoric conditions stabilize the metastable supercooling state but do not transition it into a state of stable thermodynamic equilibrium. Leveraging the stabilizing mechanism associated with the isochoric thermodynamics of the chamber, we effectively eliminate the probability of nucleation in the supercooled matter and solution within the polymer bag at the chosen storage temperature.

It should be noted that, if the solution in the polymer bag or the organ were to freeze, the system would enter a state of stable thermodynamic equilibrium, leading to an increase in pressure within the isochoric chamber proportional to the percentage of ice formed, as demonstrated in [34,43]. As a result, pressure measurements can serve as a reliable indicator for detecting ice formation in the isochoric chamber and potential preservation failure if such an event were to occur.

The isochoric chamber depicted in Figure 2 is equipped with temperature sensors to monitor the fluid temperature inside it. Additionally, the chamber is designed to monitor the internal pressure using pressure sensors. It is sealed tightly and engineered to withstand a specified internal pressure. Monitoring the pressure serves as a means to identify freezing events within the chamber. This measurement is utilized in a feedback loop to terminate the cooling process and initiate warming if an increase in pressure is detected. As mentioned in the previous paragraph, this characteristic is an important aspect of the isochoric supercooling cold storage system, preventing the freezing-induced damage to the stored biological matter when an ice nucleation event is detected by the pressure transducer.

### 2.2. Isochoric Supercooling Preservation Chamber

The core of the isochoric supercooling system is the isochoric chamber, which has a spacious internal volume of 11 L. This volume was chosen to make the chamber suitable for storing various amounts and sizes of preserved materials under isochoric conditions (as shown in Figure 2). The chamber walls are designed to withstand an internal pressure of 1.5 MPa.

The isochoric chamber is a cylindrical structure with a sealed lid. Figure 3 illustrates both the top view (A) and side view (B) of the open isochoric chamber. The main component of the chamber is a section of a DN300 pipe made of austenitic stainless steel AISI 321 W1.4541, supplied by ITALINOX ROMANIA SRL. It has an internal diameter of 303 mm and a height of 150 mm. The bottom of the pipe is equipped with a 13 mm thick plate made of the same austenitic stainless steel AISI 321 W1.4541, which is welded in place. On the top of the tubular isochoric chamber, there is a 4 × 2 mm indentation designed to accommodate a rubber O-ring. The specific O-ring used is a silicone rubber NBR70 O-ring from Dichtomatik–Freudenberg FST GmbH, Germany, capable of operating within temperatures ranging from −30 °C to +100 °C. Additionally, an 11 mm high and 10 mm wide ring is welded to the interior wall of the chamber, positioned 10 mm below the top. This ring serves as a slot for a second O-ring, a 5 mm silicone rubber NBR70 O-ring, also suitable for temperatures ranging from −30 °C to +100 °C. The outer ring has twelve holes that accommodate M16 × 90 mm screws, allowing for secure fastening of the isochoric chamber. To provide stability and elevation, the isochoric chamber is supported by four support legs made of austenitic stainless steel AISI 321 W1.4541, with a diameter of Ø14 × 2 mm and a height of 30 mm. These support legs are welded as depicted in Figure 3. The purpose of these support legs is to create sufficient space for completely surrounding the isochoric chamber with the cooling bath fluid, which is a mixture of 50% water and 50% ethylene glycol.

The lid of the isochoric chamber is constructed using a 19 mm thick plate made of austenitic stainless steel AISI 321 W1.4541, sourced from ITALINOX ROMANIA SRL. Figure 4 provides visual representations of both the top and bottom views of the lid.

The isochoric chamber lid, as depicted in Figure 4, is equipped with a pressure transducer and two temperature sensors. These sensors extend to different heights inside the chamber. Additionally, the isochoric chamber is designed with a solution filling port and an overflow port, as shown in Figure 4. The temperature sensors used are TR10 resistance temperature detectors (RTD) manufactured by Endress+Hauser AG in Switzerland. These sensors are capable of measuring temperatures within a range of −200 °C to +600 °C, even in systems under pressures of up to 7.5 MPa. They provide temperature readings at two distinct locations within the isochoric chamber, namely, at the top and bottom. The pressure transducer employed is the Cerabar PMC11 gauge pressure transducer, also manufactured by Endress+Hauser AG in Switzerland (as seen in Figure 4B). It utilizes a capacitive, oil-free ceramic sensor and is capable of measuring gauge pressures ranging from 0.04 MPa to 4 MPa.

Figure 5A provides a depiction of the assembled isochoric chamber, wherein the isochoric chamber lid is securely fastened to the chamber body using twelve M16 × 90 mm screws. The insert within the figure illustrates the arrangement of the O-rings. In Figure 5B, the assembled lid is shown, featuring the thermal sensors (thermistors) and the pressure transducer. Additionally, both Figure 4 and Figure 5A highlight the presence of an overflow and air-purging valve. The inclusion of an overflow and air-purging valve is of utmost importance within the system, as the isochoric concept is compromised when free air is present. The top surface of the cover plate is equipped with a filling valve, as depicted in Figure 4 and Figure 5A. This valve is a Badotherm instrument needle valve (Badotherm BDTV910), designed with a maximum working pressure of 41.3 MPa at 38 °C. It possesses a non-rotatable conical tip, ensuring precise alignment, and all components are crafted from AISI 316(L) low-carbon stainless steel. The valve connectors are 1/2″NPT-m. It can withstand a maximum pressure of 41.3 MPa at 38 °C and a maximum temperature of 240 °C. Figure 5B showcases the RTD thermistors situated on the interior part of the chamber. These two thermistors have varying immersion lengths, positioned at 150 mm and 50 mm from the bottom of the isochoric chamber lid. To prevent the structures protruding from the top surface of the chamber from acting as fins, thermal pipe insulation is applied to provide insulation.

### 2.3. Refrigeration System

The isochoric chamber is placed within a temperature-controlled fluid in the cooling bath, which is specifically designed to maintain the isochoric chamber at the desired temperature (as depicted in Figure 6). The cooling bath comprises a stainless-steel container with a diameter of 45 cm and a height of 28 cm, as illustrated in Figure 6A. An outer insulation measuring 19 mm (with a thermal conductivity value of λ ≤ 0.036 W/m·K) is wrapped around the cooling bath. The bath is equipped with three ports: inlet flow, outlet flow, and overflow return. In Figure 6B, the assembled isochoric chamber can be seen immersed in the cooling fluid within the cooling bath.

The cooling bath fluid itself is a mixture of 50% water and 50% ethylene glycol. The system utilizes continuous recirculation cooling, in which cooled fluid from a refrigeration system is directed into the cooling bath via the inlet flow port and subsequently exits the bath through the outlet flow port. To prevent any spillage during the temperature control process, an overflow port is included within the system.

The cooling bath fluid for the isochoric chamber is cooled within the evaporator chamber of a conventional refrigerator system. It is important to note that the refrigeration of the cooling bath fluid can be accomplished using various commercially available recirculating chilling baths, such as the “PolyScience Refrigerated Circulating Baths” from Niles, IL, USA. These baths are capable of accommodating volumes of up to 75 L and have built-in controls to maintain a constant temperature of the fluid. It is common for the cooling fluids used in these baths to be a 50% water and 50% ethylene glycol mixture. However, for our convenience, we have developed our own refrigeration system based on the design of commercial systems. Figure 7 illustrates the refrigeration system employed for the isochoric chamber cooling bath fluid.

The core component of our refrigeration system is a mechanical vapor compression cooling unit with an air-cooled condenser (Tecumseh AE4440 HR, PS30 bar, TS125 °C, from Tecumseh Products Company LLC, Ann Arbor, MI, USA), which operates using a hermetic piston compressor (Tecumseh AE-8036-BR, from Tecumseh Products Company LLC, Ann Arbor, MI, USA). The refrigerant used in our system is R404A. The refrigerator’s evaporator consists of copper coils located along the walls of the evaporator chamber. To prevent stratification within the evaporator cooling bath, we utilize a recirculation pump (DAB EVOSTA2, 40-70/130 1″, 3.6 m^3^/h, from Dab Pumps S.p.A., Mestrino (PD), Italy) with a maximum head of 7 mH_2_O.

### 2.4. Control Hardware and Software

Figure 7 presents a schematic diagram of the complete isochoric supercooling preservation system. The system operates by utilizing the “hydraulic circuits” unit, which is under the control of a dedicated control system. This unit operates based on inputs from temperature transducers located in both the isochoric chamber and the isochoric chamber cooling bath. The purpose of this control system is to ensure a constant and desired temperature within the isochoric chamber. Additionally, the control system is linked to the pressure transducer, which serves as a safety feature.

An essential aspect of the isochoric supercooling preservation system is the inclusion of measures to prevent complete freezing of the biological matter when ice nucleation, manifested as an increase in pressure, is detected. The isochoric chamber is designed to withstand a maximum pressure of 1.5 MPa. The automation and hydraulic systems are configured to establish a safety threshold for the chamber pressure, typically set between 0.2 to 0.6 MPa. When this threshold is reached, as detected by the pressure sensor, the automation system halts the cooling process. It sends a signal to the 3-way valve (marked as M in Figure 8) to cease the flow of chilled fluid from the evaporator’s cooling bath into the isochoric chamber cooling bath. Simultaneously, the recirculating pump (marked as Pc in Figure 8) initiates the recirculation of the cooling bath fluid. During the recirculation, the fluid in the isochoric chamber’s bath undergoes heat exchange with the surrounding environment through natural and forced convection, causing its temperature to rise. This interruption of the freezing process limits the increase in internal pressure within the isochoric chamber. Enhancements to the heating process could be achieved by the addition of an electric heating element inside the cooling chamber.

The temperature of the evaporator cooling bath can be adjusted to any desired value. It is set according to the desired temperature within the isochoric chamber. Typically, the temperature in the evaporator chamber is maintained around 1–1.5 °C lower than the temperature in the isochoric chamber’s cooling bath. Regardless of the temperature in the evaporator bath, the automation system ensures that the temperature in the isochoric chamber’s cooling bath remains constant. This is achieved through the use of the 3-way valve (marked as M in Figure 8) and the recirculation pump (marked as Pc in Figure 8). Closing the 3-way valve does not impact the temperature in the evaporator bath.

The control system consists of three main components: (1) the programmable logic controller (PLC), (2) power sources, fuses, and relays, and (3) the human–machine interface (HMI). The control panel enables users to remotely control the process through the internet using VNC and has the capability to store all recorded data from the temperature and pressure sensors on a memory stick or SD card. The input data is obtained from the pressure transducer, the two RTD temperature sensors in the isochoric chamber, and the RTD temperature sensor in the isochoric chamber bath. The temperature control system for the isochoric chamber (Dixell, XR20XC, Copeland Europe GmbH, Aachen, Germany) receives inputs from the temperature and pressure sensors attached to the isochoric chamber (as shown in Figure 4 and Figure 5). The control system regulates the flow through the isochoric chamber cooling bath (as shown in Figure 6 and Figure 7) to maintain the desired temperature inside the isochoric chamber. When the pressure transducer detects an increase in pressure, indicating the beginning of freezing, the controller stops the coolant flow from the evaporator to the cooling chamber.

The system operates based on the following principle. The automation panel consists of a CPU (Siemens SIMATIC S7-1200 model CPU1214C, Siemens AG, München, Germania) with the capability to handle 14 digital inputs at 24 V DC, 10 digital outputs at 24 V DC/0.5 A, and 2 analog inputs ranging from 0–10 V. To extend the digital or analog I/O capacity, three additional modules are installed on the right side of the CPU. The first two modules are Siemens SM1231 devices, where one receives signals from the pressure transducer and the other from the RTD temperature sensors. The third module, a Siemens SM1232, controls the 3-way valve responsible for maintaining a constant temperature in the isochoric chamber cooling bath.

For power supply, all components utilize a 24 V switched-mode power supply unit from Siemens (Siemens, PM1207, Siemens AG, München, Germania). The human–machine interface is achieved through a Siemens Simatic HMI TP700 Comfort panel, featuring a 7″ widescreen-TFT-display with touch operation, 16 million colors, PROFINET interface, MPI/PROFIBUS DP interface, and 12 MB user memory. This control panel enables users to remotely control the process via the internet using VNC (RealVNC^®^ Limited, Cambridge, UK) and has the capability to store all information on a memory stick or SD card. The human–machine interface includes a touchscreen display, allowing users to observe and interact with all the measured parameters in the system.

The automation box receives analog inputs from two RTD temperature sensors in the isochoric chamber, one RTD temperature sensor from the fluid in the isochoric chamber fluid bath, and one absolute pressure gauge in the isochoric chamber. The digital outputs consist of one relay (K1) for the refrigeration system command and another relay (K2) for the cooling fluid recirculation pump (Pc) command in the isochoric chamber cooling bath. The analog output controls the 3-way regulating valve. Figure 9 provides a photograph of the electronics inside the electrical and automation panel, and a detailed electric circuit can be found in the Appendix A.

In order to document all experiments, we utilized an IP wireless smart camera (PNI-UK LTD, model IP930W 1080P, Luton, UK) capable of capturing full HD 1080p video with infrared and night-vision capabilities. The camera was connected to the system through a 4G LTE Cat6 wireless router (Huawei, model B525s-23a, Shenzhen, China).

### 2.5. Experimental Protocol

This section outlines the assembly protocol and highlights important steps for setting up the experiment. Errors during these steps can result in the failure of the experiments due to deviating from isochoric conditions.

The first step is to fill the isochoric chamber cooling bath and the evaporator chamber with a solution that can maintain the desired preservation temperature. In this study, we used a solution consisting of 50% by volume ethylene glycol and 50% by volume water. In typical experiments, this solution is pre-cooled to 4 °C before the experiment begins. Since the thermal mass is significant, pre-cooling the system to the desired initial temperature can affect the outcome of the biological matter preservation protocol.

The isochoric chamber is prepared as follows: At the beginning of the experiment, the correct placement of the O-rings and the secure positioning of all the sensors are verified. The interior of the isochoric chamber and the top cover are cleaned to ensure there are no residues on the walls and transducers. Then, the open isochoric chamber is filled with a solution that has a composition designed to prevent freezing at the preservation temperatures. It is possible for air bubbles to appear in the solution. Previous studies have demonstrated the importance of minimizing and preferably eliminating the volume of free air in the system. Therefore, it is recommended to fill the isochoric chamber slowly and allow it to overflow.

The biological matter to be preserved is placed inside LDPE bags, which are then filled with an organ preservation solution. Special attention is given to ensuring that all air is removed from the bags before they are sealed. Submerging the bags in the isochoric chamber will lead to additional overflow of the fluid within the isochoric chamber.

The next step involves placing the lid onto the O-rings with caution, while keeping the overflow valve on the top lid open. This ensures that the isochoric chamber is fully filled with fluid. The lid is then securely fastened to the chamber body using twelve screws and nuts, as depicted in Figure 4B.

As mentioned earlier, it is crucial to minimize and preferably eliminate free air from the isochoric system. To achieve this, the overflow connection is opened after securing the lid to the chamber. Then, fluid is gradually added through the solution flowing port (Figure 5A) until it begins to flow out from the overflow port. The valves for both ports are then closed, and the screws are tightened further. During this stage, it is important to monitor the pressure transducer, as the final tightening process can be used to set the initial pressure for the experiments. The initial pressure has an impact on the thermodynamics of the process.

There are two essential elements that need to be carefully controlled in isochoric studies: the elimination of free air from the system and the initial pressure within the system. The next step in the experiment involves submerging the isochoric reactor in the cooling bath of the isochoric chamber (Figure 6B). The temperature and pressure of the isochoric chamber are continuously monitored and controlled throughout the experiment.

Upon completion of the designated storage period, the isochoric system is heated to a temperature above freezing and opened.

## 3. Results and Discussion

### 3.1. Control System in Response to Ice Nucleation

A key aspect of the isochoric storage technology introduced in this paper is the control mechanism implemented to safeguard against organ damage caused by unexpected and random ice nucleation. This control mechanism involves a combination of pressure detection and chamber warming. This experiment aimed to demonstrate the performance of the control system in response to ice nucleation and the subsequent pressure increase. The isochoric chamber was filled with steam-distilled water, following the previously described protocol. The cooling fluid in the cooling chamber was set to maintain a temperature of −2 °C, while the evaporator chamber’s fluid temperature was set to −3 °C. The chamber itself is designed to withstand a pressure of 1.5 MPa. The control system was programmed to cease cooling and initiate warming recirculation when the pressure transducer recorded a pressure of 0.2 MPa. Figure 10 illustrates the pressure history in the water-filled isochoric chamber. It reveals that the pressure remained constant from 14:18 h to 21:18 h, indicating a supercooled state of the water for a duration of seven hours. Subsequently, a sudden increase in pressure was observed, indicating ice nucleation. The control system promptly responded at 0.2 MPa, activating the three-way valve, M, to halt the flow of cooling fluid from the evaporator chamber into the cooling chamber and initiating the recirculating pump, Pc, in the cooling chamber (as shown in Figure 7). This led to the warming of the cooling fluid in the cooling chamber and, consequently, the warming of the isochoric chamber. Figure 10 demonstrates that this warming process slowed down the freezing, resulting in a deceleration of the pressure increase. Eventually, the warming initiated the melting of the formed ice, leading to a reduction in the amount of ice within the system. It should be noted that, based on fundamental thermodynamic principles, in an isochoric two-phase system, temperature and pressure are uniquely correlated by the liquidus line on the phase diagram. Therefore, the pressure measurements can be directly translated into temperature. The warming of the cooling fluid, in the absence of fluid flow between the evaporator chamber and the isochoric bath cooling chamber, occurs due to thermal contact between the recirculating fluid and the surroundings. Faster melting could be achieved by implementing active heating within the isochoric chamber or by directly heating the chamber walls. It is important to emphasize that the chamber is designed to withstand pressures up to 1.5 MPa, ensuring that the return to the original pressure is not due to leakage.

The experimental results presented in this section provide compelling evidence of the efficacy and practicality of the pressure-related control mechanism introduced in this study. The control mechanism, which combines pressure detection and chamber warming, proves to be instrumental in preventing organ damage due to unexpected and random ice nucleation.

The experiments demonstrate that the pressure sensors accurately detect any increase in pressure within the isochoric chamber, signaling the occurrence of ice formation. This prompt detection allows for immediate action to halt further ice nucleation and protect the organ from potential harm.

Furthermore, the application of controlled chamber warming in response to pressure indications effectively reverses the freezing process, restoring the organ to its supercooled state. This preventive measure proves crucial in maintaining the integrity and viability of the preserved organ.

Overall, the experimental results validate the significance of the pressure-related control mechanism in ensuring the success of isochoric storage technology for organ preservation. The mechanism’s ability to respond swiftly and accurately to ice nucleation events showcases its potential as a reliable and indispensable component of the preservation process. This study’s findings undoubtedly contribute to advancing organ preservation methods and hold promise for improving the outcomes of transplantation procedures in the future.

### 3.2. Experiments on Ice Nucleation in a Sham Study with a Non-Viable Pig Liver in a Cold Storage Solution in a Polymer Bag

Static cold storage is a common technique used to extend the preservation period of biological organs for transplantation by reducing their metabolism through lower temperatures. Conventional static cold storage is typically limited to temperatures at around 4 °C, as lower temperatures below the freezing point (0 °C) can indeed lead to ice formation, which is harmful to the organ. The formation of ice during subfreezing temperatures can cause severe damage to the organ’s cellular structure and functionality, making it unsuitable for transplantation. This limitation is a significant challenge in conventional cold storage methods, as they aim to strike a balance between reducing metabolism and avoiding ice formation. The technology introduced in the paper overcomes this limitation by preserving the organ at subfreezing temperatures in a supercooled state. By maintaining the organ in a supercooled liquid state below the freezing point, the detrimental effects of ice formation can be circumvented, extending the preservation period without compromising the organ’s viability for transplantation.

Supercooling is a metastable state where ice nucleation becomes a random and stochastic process, making it challenging to avoid ice formation at subfreezing temperatures. Ice formation can be induced by various mechanisms, including homogeneous nucleation, surface nucleation, and nucleation from ice nuclei in the system’s bulk. In this study, the isochoric system effectively addresses these ice nucleation mechanisms. Isochoric conditions significantly reduce homogeneous nucleation temperature [42], while the use of a high-concentration “isochoric chamber fluid” and polymer bags eliminates surface nucleation. Previous studies suggest that confinement in an isochoric chamber stabilizes the supercooled state, mitigating the effect of ice nuclei in the bulk of the preservation solution and the organ [39]. By leveraging this stabilizing mechanism, the technology eliminates the probability of nucleation in the supercooled organ and solution within the polymer bag at the chosen storage temperature. However, it is essential to note that isochoric conditions do not transition the supercooled state into stable thermodynamic equilibrium. If the solution in the polymer bag or the organ were to freeze, the system would enter a state of stable thermodynamic equilibrium, leading to an increase in pressure within the isochoric chamber proportional to the percentage of ice formed, as demonstrated in [34,43].

In the isochoric system during supercooling storage, measuring the pressure becomes a crucial method for detecting ice formation and potential preservation failure. When the organ and solution are in a supercooled state, the system remains metastable, but if ice formation occurs due to any unforeseen event or nucleation process, it transitions to a stable thermodynamic equilibrium. As ice forms, it takes up more space compared to the supercooled liquid, leading to an increase in pressure within the isochoric chamber. By continuously monitoring the pressure inside the isochoric chamber, any sudden rise in pressure serves as an early warning sign of ice nucleation. This timely detection allows for immediate action to prevent further ice formation and potential damage to the organ. The pressure measurement serves as a reliable indicator of the system’s stability and the effectiveness of the isochoric storage technology in avoiding ice formation during supercooling storage. It enables researchers to verify that the organ remains in the desired supercooled state, free from ice nucleation, and ensures the success of the preservation process.

The primary objective of the experiment is to validate that the designed isochoric system effectively avoids freezing under the specified conditions. To accomplish this, sham experiments were conducted using whole non-viable pig livers from a local food vendor submerged in physiological saline. These experiments were designed to assess the system’s capability to sustain supercooling without any ice nucleation occurring for extended periods. It is essential to highlight that the experiment’s main focus is to demonstrate the technology’s ability to prevent ice formation during isochoric storage. It is not intended to study the survival of a live liver during isochoric storage. By using non-viable pig livers and physiological saline, we verify the effectiveness of the isochoric system in maintaining the supercooled state without the potential complications associated with live organ preservation. Through this experiment, we aim to provide evidence supporting the reliability and efficacy of the isochoric storage technology, further solidifying its potential application in the field of organ preservation for transplantation.

Three non-viable pig livers, weighing between 1.2 kg and 1.5 kg, were obtained from a local food vendor. They were individually placed inside 2 L low-density polyethylene (LDPE) storage bags with zip-lock seals. Each bag was filled with a 0.9% NaCl (physiological saline) solution, ensuring that no air bubbles are trapped, and then securely sealed. Within the isochoric chamber, a carefully poured isochoric chamber fluid solution of 3M NaCl was introduced, ensuring no air bubbles were trapped. The 3M NaCl solution has a freezing temperature of approximately −11 °C. This specific composition was selected to prevent freezing at the experimental temperature of −2 °C, which was the designated temperature for the isochoric supercooling preservation experiment. The pouring process, described in detail in Section 2.5 of the Materials and Methods, involved carefully adding the 3M NaCl solution around the plastic bag inside the isochoric chamber. Figure 11A visually presents the liver contained within the plastic bag placed inside the isochoric chamber. To address the liver’s tendency to float in the 3M NaCl solution, a polystyrene plastic lid was positioned between the liver and the chamber’s lid. The LDEP storage bag made contact with the surrounding fluid in the isochoric chamber, except at the points where it touched the plastic lid. Following the instructions provided in Section 2.5, the isochoric chamber was sealed, ensuring a secure closure. Subsequently, the sealed chamber was immersed in the isochoric chamber cooling bath, as outlined in Section 2.5.

In analyzing the results, it is beneficial to include data on the properties of the materials present in the isochoric chamber. The bulk moduli of the relevant materials are as follows: air, 10–4 GPa; water, 2.2 GPa; ice, 10 GPa; polyethylene (plastic bag), 0.3 GPa; polystyrene (lid), 1.8–2.4 GPa; steel, 160 GPa. Examining these values, we can observe that the plastic bag and the polystyrene lid have minimal impact on the compressibility of the materials within the isochoric chamber. The bulk modulus of the polystyrene lid is comparable to that of water. The LDEP bag has a lower bulk modulus than water, but due to its thickness of 30 µm, the volume it occupies is negligible compared to the 11 L volume of water. Furthermore, the bulk modulus of LDEP is three orders of magnitude greater than that of air. The inhibitory effect of isochoric conditions on random nucleation is attributed to the propagation of pressure information regarding nucleation events to the rigid walls of the isochoric chamber, leading to the Le Chatelier effect. Although the speed of sound in water is 1234.8 m/s and the speed of sound in LDEP is only 540 m/s, the thickness of the plastic bag is five orders of magnitude smaller than the typical dimensions of the chamber. Therefore, it is highly unlikely that the plastic bag will have a detrimental effect on the isochoric mechanism that maintains the stability of metastable supercooling in the plastic bag.

The temperature within the isochoric chamber was set to −2 °C, and real-time monitoring of temperature and pressure was conducted. It should be noted that the freezing temperature of physiological saline is −0.54 °C, while the freezing temperature of 3M saline is approximately −11 °C. Consequently, the liver enclosed within the low-density polyethylene bag was in a supercooled state, and the solution surrounding the bag was not expected to freeze.

A total of one 24 h experiment and two 48 h experiments were performed. None of these experiments exhibited ice nucleation, and the livers remained in a supercooled state throughout the duration of the experiments. It is important to note that the ice nucleation experiments were conducted differently from the experiments described in this section. In the ice nucleation experiments, the isochoric chamber was filled with pure water, and there was no attempt to inhibit nucleation from the chamber walls.

The results from a repeated 48 h experiment are illustrated in Figure 12 and Figure 13. Figure 12 depicts the temperature within the cooling bath and at two locations inside the isochoric chamber as a function of time. It can be observed that the temperature remained constant at −2 °C throughout the entire experiment.

Figure 13 displays the pressure inside the isochoric chamber during the 48 h storage period. It can be observed that there was no increase in pressure throughout the entire 48 h experiment. The absence of pressure elevation was consistent across all three experiments conducted.

Importantly, previous research [34,43] has indicated that even a 1% by volume freezing within an isochoric system would result in a pressure increase from 0.1 MPa to 1 MPa. As depicted in Figure 10, our device would have detected such a pressure change. Therefore, these findings provide evidence that there was no ice nucleation in the supercooled liver contained within the plastic LDPE bag, suggesting that the system introduced in this paper could successfully achieve long-term isochoric supercooling of organs, even those as large as the pig liver.

In summary, the experiments conducted in this study showcase the significance of pressure monitoring during isochoric supercooling preservation. The first experiment highlights the value of pressure detection as it can activate a temperature control mechanism, effectively reducing the likelihood of ice formation within the organ. This control mechanism safeguards the organ from freezing damage, ensuring its integrity and viability for transplantation.

In the second experiment, the researchers demonstrate the successful preservation of the liver in a supercooled state for 48 h without any evidence of ice nucleation detected by an increase in pressure. This outcome reinforces the effectiveness of the isochoric system in avoiding ice formation during the preservation process.

Overall, the study’s findings provide strong evidence supporting the reliability and practicality of pressure monitoring as a critical component of the isochoric supercooling preservation technique. The technology’s ability to maintain the organ in a supercooled state and prevent ice nucleation holds promise for improving organ preservation methods, ultimately contributing to better outcomes in transplantation procedures.

It is essential to emphasize that the novel aspect of the isochoric supercooling chamber lies in the isochoric chamber itself and the isochoric chamber container, which is illustrated as the last element in Figure 8. All the other components can be replaced by programmable chilling recirculators, which are widely available in the market. The current design can be utilized as it is in a hospital setting to extend the preservation period before organ transplantation.

However, it is worth noting that the current device was intentionally overdesigned for safety purposes in its initial prototype. To make improvements, it should be possible to substantially reduce the weight of the isochoric chamber by designing it to withstand lower pressure and replacing the metal with polycarbonate material. An important advantage of closed isochoric supercooling preservation over isobaric storage is the system’s stability to external excitations [39]. This attribute allows for convenient transportation using various modalities. With this stability, the isochoric supercooling preservation system can be effectively transported without compromising the preservation of organs, making it a promising technology for practical application.

Regarding the actual practical use of the isochoric system by clinicians, we added the cautionary statement about avoiding air in the bag to make users aware of this potential issue. However, clinicians in our group, who possess substantial experience in organ transplantation, have evaluated the loading process into the plastic bag and found it to be very similar to loading an organ into a conventional storage bag for cold storage transportation. The design of the isochoric chamber loading ensures the elimination of any possibility of air in the chamber.

We would like to highlight the ease of use of cold storage isochoric supercooling preservation compared to machine perfusion, which makes it an attractive attribute of the technology. One of the main reasons for transitioning to the more complex machine perfusion technique is its capability to provide longer preservation times than conventional cold storage. Nevertheless, with isochoric supercooling storage, it should be feasible to achieve preservation times comparable to machine perfusion while employing much simpler procedures. This suggests that the isochoric supercooling method has the potential to offer a practical alternative for clinicians, combining effective preservation with user-friendly operation.

## 4. Conclusions

In conclusion, this technical study introduces a novel isochoric supercooling preservation system tailored for the unfrozen preservation of large biological organs at subfreezing temperatures. The experiments conducted on non-viable pig livers obtained from a local food vendor serve as evidence, demonstrating that organs as large as the pig liver can be effectively maintained in a supercooled state for extended periods, up to 48 h, without any ice nucleation occurring within the system.

This technology offers exciting potential for long-term organ preservation, eliminating the requirement for cryoprotectant additives. The ability to preserve organs without freezing opens up new possibilities for organ transplantation procedures, potentially enhancing outcomes and extending the viability of donor organs. However, final evidence that these preserved organs can be successfully transplanted is still lacking.

## Figures and Tables

**Figure 1 bioengineering-10-00934-f001:**
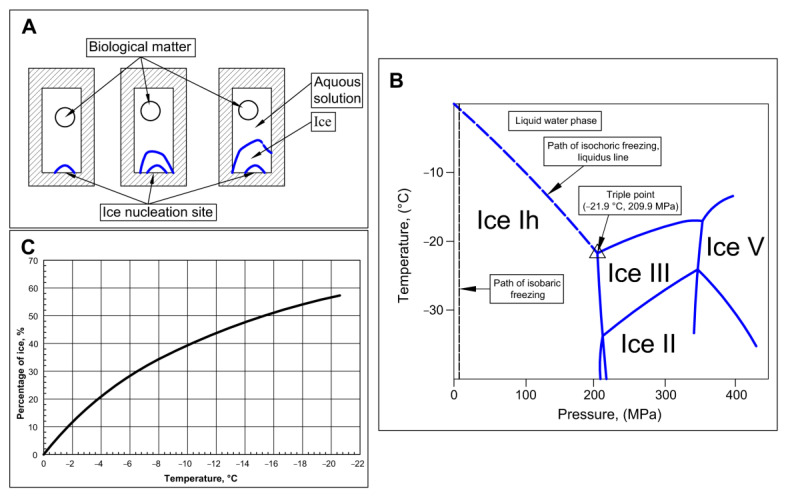
Schematic of isochoric (constant volume) freezing. (**A**) schematic of the process of freezing in an isochoric chamber. (**B**) The process of freezing in an isochoric system and an isobaric system depicted on an ice water phase diagram (**C**) percentage of ice in an isochoric system as a function of temperature.

**Figure 2 bioengineering-10-00934-f002:**
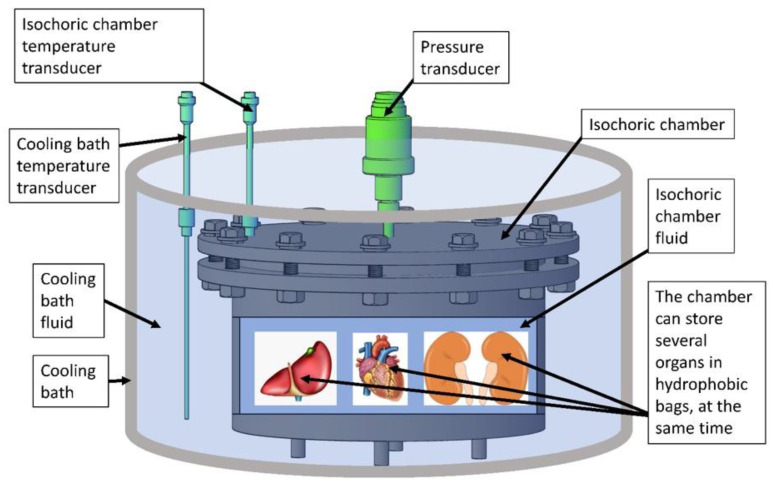
The schematic illustrates the isochoric supercooling system, which comprises a sealed “isochoric chamber” filled with isochoric chamber fluid. The chamber provides space for storing multiple organs or biological tissues, enclosed in thin plastic hydrophobic bags. These bags facilitate heat and pressure transfer but prevent mass transfer. The pressure inside the isochoric chamber is monitored by a pressure transducer, while temperature sensors keep track of the isochoric chamber fluid’s temperature. The isochoric chamber is immersed in a “cooling bath” containing a cooling bath fluid, which maintains the desired preservation temperature. A temperature transducer, submerged in the cooling bath fluid, measures the temperature of the bath fluid.

**Figure 3 bioengineering-10-00934-f003:**
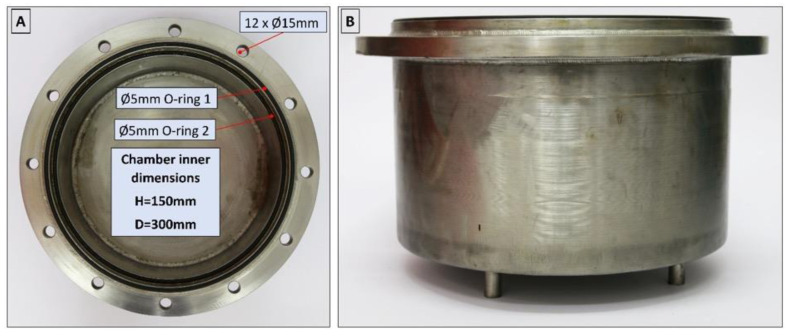
Top (**A**) and side view (**B**) of the open isochoric chamber.

**Figure 4 bioengineering-10-00934-f004:**
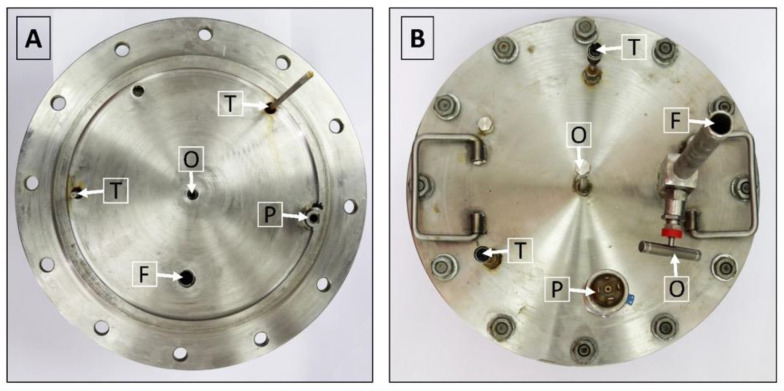
(**A**) View of the isochoric chamber lid from the bottom. (**B**) View of the isochoric chamber lid from the top. O—overflow port, T—thermal transducers, P—pressure transducer, F—filling port.

**Figure 5 bioengineering-10-00934-f005:**
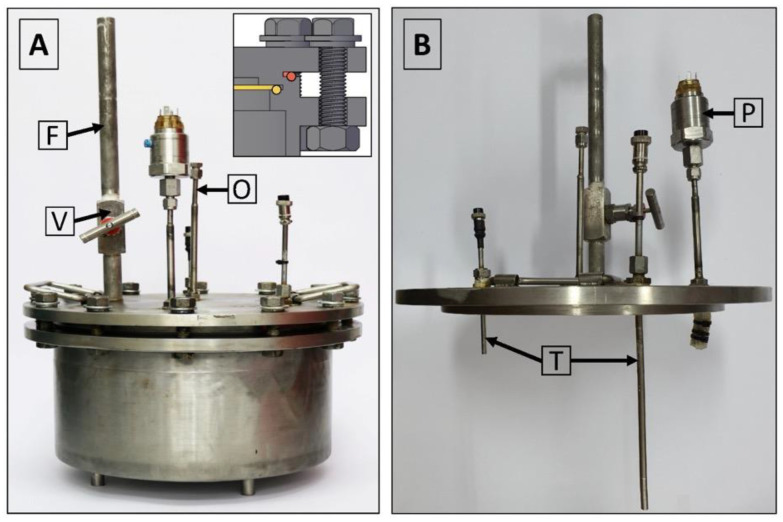
(**A**) Assembled isochoric chamber, (**B**) assembled lid. O—overflow port, T—thermal transducers, P—pressure transducer, F—filling port, V shut-off valve.

**Figure 6 bioengineering-10-00934-f006:**
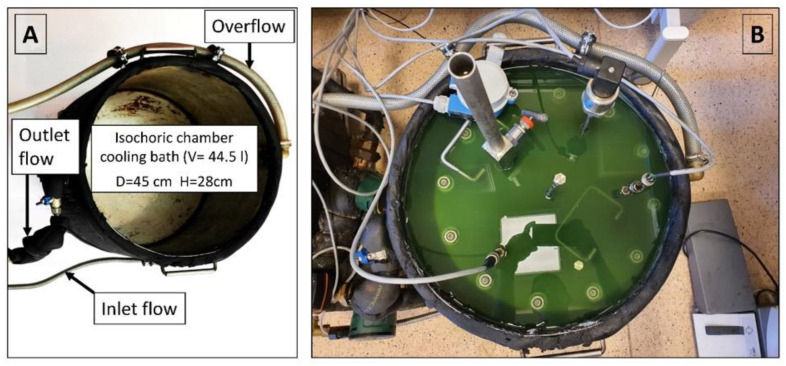
(**A**) Isochoric chamber cooling bath. (**B**) Isochoric chamber immersed in the cooling bath fluid.

**Figure 7 bioengineering-10-00934-f007:**
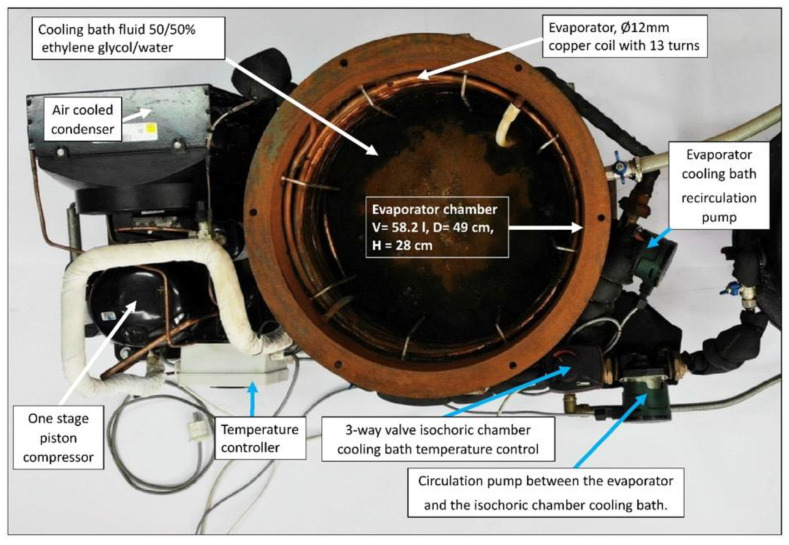
The evaporator chamber in which the isochoric chamber cooling bath fluid is cooled by the evaporator of a homemade compression refrigeration system.

**Figure 8 bioengineering-10-00934-f008:**
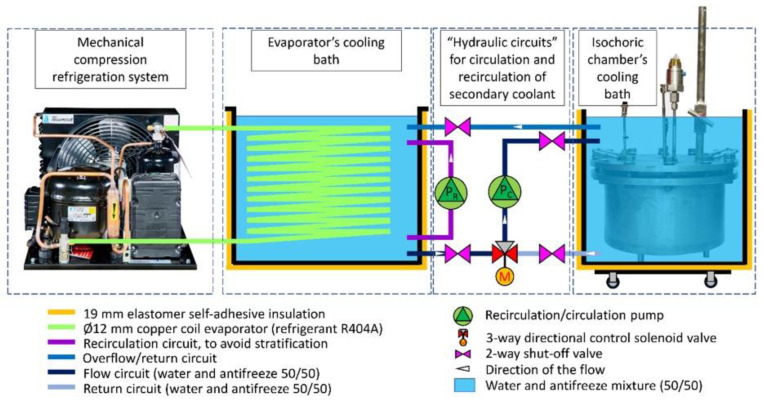
Schematic of all the components of the isochoric supercooling refrigerator. The highlighted “hydraulic circuits” elements are controlled by the control system. Pump Pr recirculates the cooling fluid in the evaporator’s cooling bath to eliminate stratification; pump Pc circulates the cooling fluid between the evaporator’s cooling bath and the isochoric chamber’s cooling bath or recirculates the fluid in the isochoric chamber’s bath; the 3-way directional control solenoid valve, M, is diverting the fluid between two different directions.

**Figure 9 bioengineering-10-00934-f009:**
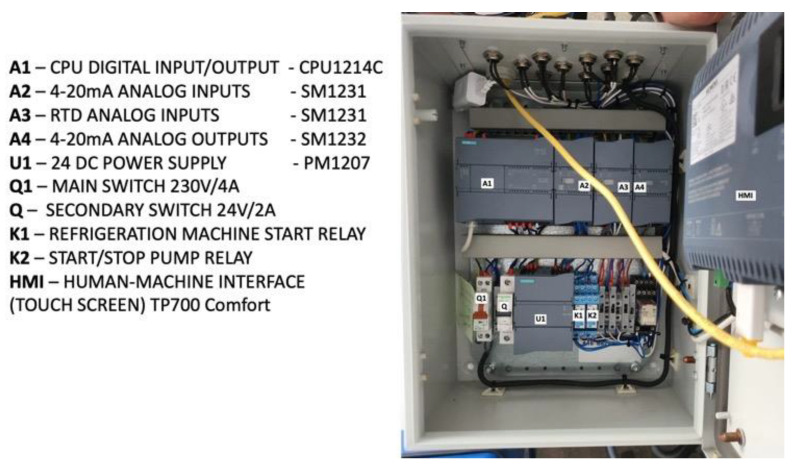
The electrical and automation panel.

**Figure 10 bioengineering-10-00934-f010:**
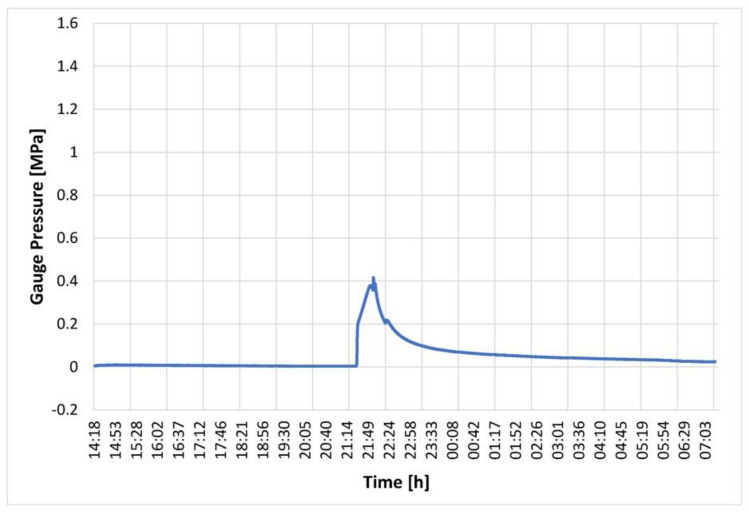
Pressure in an isochoric chamber filled with water as a function of time after the start of the experiment. Temperature control began when the pressure reached 0.2 MPa.

**Figure 11 bioengineering-10-00934-f011:**
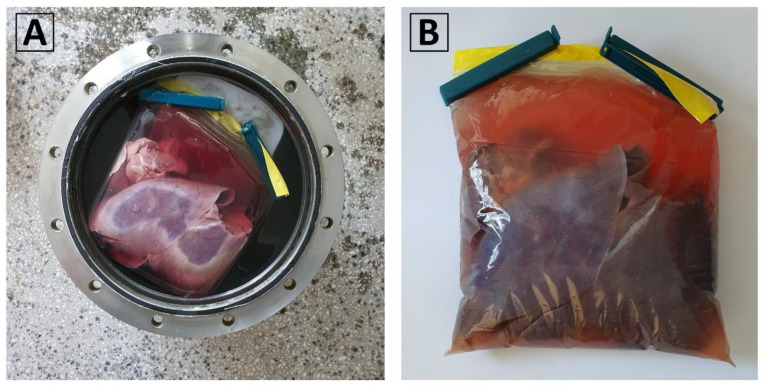
(**A**) The pig liver inside the 2 L LDPE plastic bag in the isochoric chamber filled with 3M NaCl. (**B**) The liver in the sealed plastic bag.

**Figure 12 bioengineering-10-00934-f012:**
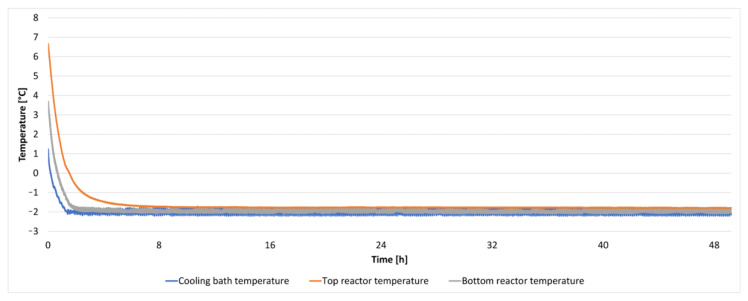
The temperature inside the cooling bath and the isochoric chamber during a 48 h isochoric supercooling experiment.

**Figure 13 bioengineering-10-00934-f013:**
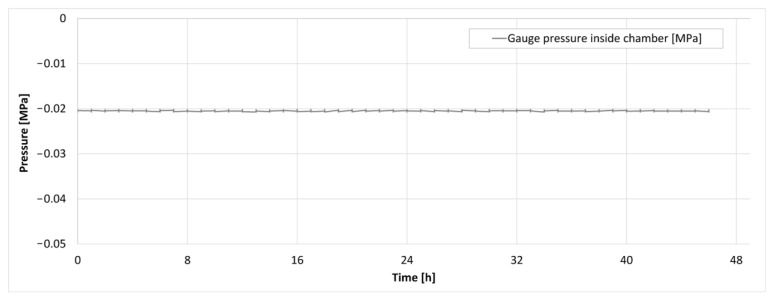
The pressure trace inside the isochoric chamber during the supercooling experiment.

## Data Availability

The data that support the findings of this study are available from the corresponding author upon reasonable request.

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
