# Peer review of "Isochoric Supercooling Organ Preservation System"

_bioengineering, 2023, doi:10.3390/bioengineering10080934_

Round 1

Reviewer 1 Report

The article presents a significant advance in organ preservation in pig livers. The methods and procedures are well described. If the engineering aspects of the paper can be validated by appropriate experts, I highly recommend publication of this paper.

Reviewer 2 Report

I have read with great interest the manuscript entitled "Isochoric Supercooling Organ Preservation System", submitted to Bioengineering. In this original article, the authors describe a novel organ preservation system based on isochoric supercooling. The manuscript is well-written, and the topic of clinical interest within the field of transplantation. Nevertheless, there are a few concerns described below.

MAJOR COMMENTS

- It is understandable that this is a technical paper; thus, it focuses mainly on the constructive and technical aspects of the newly developed device. Yet, the manuscript still mentions experiments proving the efficacy of the method using three pig livers. These experiments support the conclusion about the feasibility of preserving organs in a supercooled state within the system for 48 hours. Nevertheless, the amount of evidence reported for those experiments is minimal or none. It is not reported how ice nucleation was assessed in the organs. Also, how would the organs behave after this preservation? Therefore, either the experiments with the organs are described in detail, or the paper limits its discussion and conclusions to the constructive aspects of the device.

- Accordingly, the method for the pig liver experiments, including which sort of examination was performed (histological?), must be described adequately.

- In addition, because of the limitations of traditional static cold storage preservation, there is a resurgence in the interest of dynamic organ preservation. Although this aspect is briefly mentioned in the Introduction of the manuscript, it is not further discussed. Authors stick to the limitations of static cold storage. Furthermore, compared to the emergent data on the advantages of machine perfusion of the liver, what are the advantages/ disadvantages of supercooling preservation? A more profound discussion about this aspect would enhance readers' interest in the manuscript and highlight the importance of the developed preservation system.

Reviewer 3 Report

The authors have submitted a technical paper introducing a prototype for isochoric supercooling for preservation of large biomaterials (e.g. organs) without the need of cryoprotectants. They have described the technical details, potential hazards and pitfalls of the technique in detail and proved the concept of their work by storing porcine livers for 48 hrs at -2°C without histological evidence of ice formation. Though theoretically very interesting, there are some aspects that might limit clinical application of this technique. 

1. Can this system be transported so that donor organs can be shipped in it? Or is only a back to base approach available?

2. The loading process of organs (inkl. elimination of air) seems a very tedious process, that is - from the viewpoint of a clinician - not easily performed/can be a potential reason for graft loss is not performed right. Is there a way to make this easier/more suitable for a hospital/facility?

3. Do you have histologic images before/after organ storage to show histology? Was there, except the absence of ice formation, noticeable storage/hypothermia induced injury? Would be nice to have a figure dedicated to this as well.

Round 2

Reviewer 2 Report

Including more information about machine perfusion technology and the possible role of supercooling in organ preservation has improved the manuscript. Nevertheless, the concern regarding the poor reporting of the pig liver experiments continues.

MAJOR COMMENTS

It still lacks information on the experiments' feasibility of preserving organs in a supercooled state within the system for 48 hours. It is not reported how ice nucleation was assessed in the organs. Therefore, either the experiments with the organs are described in detail, or the paper limits its discussion and conclusions to the constructive aspects of the device.
